# Systematic Analysis of Long Non-Coding RNA Genes in Nonalcoholic Fatty Liver Disease

**DOI:** 10.3390/ncrna8040056

**Published:** 2022-07-22

**Authors:** Mirolyuba Ilieva, James Dao, Henry E. Miller, Jens Hedelund Madsen, Alexander J. R. Bishop, Sakari Kauppinen, Shizuka Uchida

**Affiliations:** 1Center for RNA Medicine, Department of Clinical Medicine, Aalborg University, DK-2450 Copenhagen SV, Denmark; jenshm@dcm.aau.dk (J.H.M.); ska@dcm.aau.dk (S.K.); 2Bioinformatics Research Network, Atlanta, GA 30317, USA; jmsdao@gmail.com (J.D.); millerh1@livemail.uthscsa.edu (H.E.M.); 3Department of Cell Systems and Anatomy, UT Health San Antonio, San Antonio, TX 78229, USA; bishopa@uthscsa.edu; 4Greehey Children’s Cancer Research Institute, UT Health San Antonio, San Antonio, TX 78229, USA; 5May’s Cancer Center, UT Health San Antonio, San Antonio, TX 78229, USA

**Keywords:** gene expression, liver, lncRNA, NAFLD, NASH, RNA-seq

## Abstract

The largest solid organ in humans, the liver, performs a variety of functions to sustain life. When damaged, cells in the liver can regenerate themselves to maintain normal liver physiology. However, some damage is beyond repair, which necessitates liver transplantation. Increasing rates of obesity, Western diets (i.e., rich in processed carbohydrates and saturated fats), and cardiometabolic diseases are interlinked to liver diseases, including non-alcoholic fatty liver disease (NAFLD), which is a collective term to describe the excess accumulation of fat in the liver of people who drink little to no alcohol. Alarmingly, the prevalence of NAFLD extends to 25% of the world population, which calls for the urgent need to understand the disease mechanism of NAFLD. Here, we performed secondary analyses of published RNA sequencing (RNA-seq) data of NAFLD patients compared to healthy and obese individuals to identify long non-coding RNAs (lncRNAs) that may underly the disease mechanism of NAFLD. Similar to protein-coding genes, many lncRNAs are dysregulated in NAFLD patients compared to healthy and obese individuals, suggesting that understanding the functions of dysregulated lncRNAs may shed light on the pathology of NAFLD. To demonstrate the functional importance of lncRNAs in the liver, loss-of-function experiments were performed for one NAFLD-related lncRNA, *LINC01639*, which showed that it is involved in the regulation of genes related to apoptosis, TNF/TGF, cytokine signaling, and growth factors as well as genes upregulated in NAFLD. Since there is no lncRNA database focused on the liver, especially NAFLD, we built a web database, LiverDB, to further facilitate functional and mechanistic studies of hepatic lncRNAs.

## 1. Introduction

The liver is the largest solid organ in the human body, which performs over 500 essential functions related to digestion, immunity, and metabolism [1]. Liver dysfunction leads to a multitude of health risks and diseases, including liver cancer, cirrhotic liver disease, non-alcoholic fatty liver disease, and acute liver failure [2,3]. Unlike most other organs (e.g., brain, heart), the liver possesses a regenerative capacity [4]. However, upon prolonged insults (e.g., excessive consumption of alcohol, drugs), the damage to the liver becomes irreparable, necessitating liver transplantation [5]. Globally, approximately two million deaths result from liver diseases annually [6]. Thus, intensive research is conducted in identifying the causes of liver diseases.

It is now well recognized that most of the mammalian genome is transcribed as RNA, yet only a few percentages of these transcripts encode for exons of protein-coding genes [7]. All of those that do not code for proteins are collectively called non-coding RNAs (ncRNAs), which include ribosomal RNAs (rRNAs), transfer RNAs (tRNAs), microRNAs (miRNAs), and long non-coding RNAs (lncRNAs) [8]. Among these ncRNAs, lncRNAs are of particular interest to researchers in different fields because lncRNAs are regarded as missing links for understanding many cellular activities and signaling pathways. Because of their broad definition–any ncRNAs longer than 200 nucleotides–it is speculated that the number of lncRNAs well surpasses that of protein-coding genes [7,9]. Indeed, more and more lncRNAs are being discovered from next-generation sequencing (NGS), especially RNA sequencing (RNA-seq), although only limited numbers of lncRNAs are functionally studied. The amount of RNA-seq data deposited in public databases [e.g., ArrayExpress, Gene Expression Omnibus (GEO), Sequence Read Archive (SRA)] is increasing rapidly [10]. However, most of these RNA-seq data are analyzed only for protein-coding genes and not for lncRNAs. Although this is especially true for the hepatology field, recent studies indicate that the dysregulation of lncRNAs in the liver contribute to liver diseases [11,12].

Among different types of liver diseases, the incidence of non-alcoholic fatty liver disease (NAFLD) is increasing rapidly due to lack of exercise and obesity (which are two major causes of type 2 diabetes, another risk factor for NAFLD) [13,14]. NAFLD (also called “metabolic associated fatty liver disease” (MAFLD) [15]) is a collective term to describe the excess accumulation of fat in the liver of people who drink little to no alcohol [16]. It is divided into two subtypes: simple steatosis, non-alcoholic fatty liver (NAFL); and a more aggressive form, nonalcoholic steatohepatitis (NASH) [17,18]. Sadly, NAFLD occurs in about 25% of the world population [19], which highlights the urgent need to understand the mechanism of this life-threatening disease. To uncover the underlying disease mechanism of NAFLD, high-throughput methods (e.g., microarrays and RNA-seq) have been employed [20,21,22,23,24,25,26,27,28,29,30,31,32]. However, these prior studies are mainly focused on protein-coding genes. Given that lncRNAs are increasingly implicated in a variety of diseases, including liver disease and NAFLD (e.g., *GAS5* [33], *HOTAIR* [34], *LINC01260* [35], *PVT1* [36]), a secondary analysis of published high-throughput data may reveal novel insights into NAFLD. For this purpose, RNA-seq data of previous studies investigating dysregulated protein-coding genes in NAFLD patients compared to healthy donors are of great interest as unlike microarrays, RNA-seq data contain expression information about lncRNAs.

To address the above point, we conducted secondary analyses of published RNA-seq data of NAFLD patients to provide a comprehensive overview of lncRNA dysregulation in this disease context. Since protein-coding genes were already analyzed and further experimentally confirmed in the original studies, here, we omitted the detailed analysis of protein-coding genes (e.g., gene ontology and pathway analyses). Instead, our analyses focus on the expression profiling of lncRNAs, which were not provided in the original studies. By performing a systematic analysis of RNA-seq data for lncRNAs using the latest gene annotation, we uncover the differential regulation of lncRNAs in various patient samples. To confirm the findings, we conducted loss-of-function experiments of dysregulated lncRNA to supplement the lack of knowledge regarding lncRNAs in the pathogenesis of NAFLD.

## 2. Results

### 2.1. Expression Profiling of LncRNAs in NAFLD Patients Compared to Healthy Donors

The primary causes of NAFLD are dyslipidemia (abnormally high amount of lipids in the blood), insulin resistance, obesity, and type 2 diabetes [37]. All these primary causes are intertwined and are linked to other metabolic diseases (e.g., cardiovascular disease, stroke). Not surprisingly, both microarrays and RNA-seq techniques have been employed to profile transcriptomic changes in NAFLD patients compared to healthy patients, including obese individuals without NAFLD [20,21,22,23,24,25,26,27,28,29,30,31,32]. Although these studies uncovered important dysregulated signaling pathways in NAFLD patients, there is an overall lack of detailed profiling of lncRNAs in these studies. To address this lack of information, we first reanalyzed the RNA-seq data from a study which profiled protein-coding genes in patients with either subtype of NAFLD [NAFL (*n* = 15) or NASH (*n* = 16)] compared with healthy normal weight (*n* = 14) and obese individuals (*n* = 12) using liver biopsy samples [32]. The major findings in this original study are: (1) many differentially expressed protein-coding genes are identified when NAFLD patients are compared to both healthy normal weight and obese individuals; (2) these differentially expressed protein-coding genes are enriched in signaling pathways involved in lipid metabolism, immunity, extracellular matrix, and cell cycle control; and (3) there is a large overlap of differentially expressed protein-coding genes in both subtypes of NAFLD compared to healthy normal weight and obese individuals [32].

When a secondary analysis of this dataset (GEO accession number, GSE126848) was performed with a threshold of 2-fold change and false discovery rate (FDR)-adjusted *p*-value < 0.05, it was observed that over 1000 genes are differentially expressed in both subtypes of NAFLD compared to the control (i.e., both healthy normal weight and obese individuals) (Figure 1A), whereas there are few genes that are differentially expressed between NAFL and NASH as well as between healthy normal weight and obese individuals, as reported previously [32] (Figure 1B; Appendix A). Compared to protein-coding genes, the numbers of differentially expressed lncRNAs are generally lower in all conditions. As stated earlier, the original study [32] provided a detailed analysis of protein-coding genes but not for lncRNAs. Thus, we focused on lncRNAs here. When the differentially expressed lncRNAs are compared between NAFLD patients and the control individuals, 163 up- and 12 down-regulated lncRNAs are shared among all comparisons (Figure 1C; Appendix A), suggesting that some differentially expressed lncRNAs are common between two subtypes of NAFLD, as is the case for protein-coding genes [32].

### 2.2. Dysregulated LncRNAs during the Progression of NASH

More recently, an RNA-seq experiment using a large cohort of European NAFLD patients was conducted [21]. In this original study, the liver biopsy samples from 206 NAFLD patients with different fibrosis stages were compared to those from 10 healthy obese controls (GEO accession number, GSE135251). The NAFLD patients were divided into two groups based on the progression of steatohepatitis (i.e., NASH), which are denoted as early (*n* = 138) and moderate (*n* = 68). Using this new data set, we attempted to further narrow down the list of differentially expressed lncRNAs identified in the previous data set of NAFLD patients compared to healthy donors. When the same threshold (2-fold and FDR-adjusted *p*-value < 0.05) was applied to this new dataset, we observed that several hundred protein-coding and lncRNA genes are up- and down-regulated in NAFLD patients compared to the controls (healthy obese individuals), even in the case of early and moderate NASH patients, where more lncRNAs are down-regulated than up-regulated compared to protein-coding genes (Figure 2A; Appendix A). When the previously identified differentially expressed lncRNAs (i.e., 163 up- and 12 down-regulated lncRNAs) were compared within this new dataset, surprisingly, only very few lncRNAs were noted as having the same trend of up- or down-regulation (Figure 2B). This results likely reflects the difficulty of comparing RNA-seq data from different studies and possibly reflects the heterogeneity of biopsy samples [38,39]. To check whether such difficulty also exists in protein-coding genes, the same set of analyses as for lncRNAs was performed, which resulted in an analogous discrepancy of differentially expressed genes between the two studies (Appendix A).

Although the overlaps among different studies and conditions are not big, those differentially expressed genes that displayed the same trend (i.e., up-regulation in all conditions and data) should represent the most likely candidates to be validated in other laboratories and studies, regardless of experimental biases. Indeed, among 24 commonly up- and 14 down-regulated protein-coding genes, well-known markers of advanced NAFLD [e.g., aldo-keto reductase family 1 member B10 (*AKR1B10*) [40,41], cell death inducing DFFA like effector c (*CIDEC*) [42,43,44], lipoprotein lipase (LPL) [45,46], platelet derived growth factor subunit A (*PDGFA*) [47], triggering receptor expressed on myeloid cells 2 (*TREM2*) [48,49,50,51], and thymidylate synthetase (TYMS) [52] are consistently up-regulated. In the case of lncRNAs, there are four commonly up- and two down-regulated lncRNAs in all conditions and data above (Figure 3). The up-regulated lncRNAs are *AJ009632.2* (Ensembl Gene ID, ENSG00000229425), long intergenic non-protein coding RNA 1639 (*LINC01639*; ENSG00000236117), MIRLET7B host gene (*MIRLET7BHG*; ENSG00000197182), and SNAP25 antisense RNA 1 (*SNAP25-AS1*; ENSG00000227906). The down-regulated lncRNAs are *AC110995.1* (ENSG00000236120) and mitotically associated long non-coding RNA (*MANCR*; ENSG00000231298).

Among the six differentially expressed lncRNAs, *SNAP25-AS1* is shown to be up-regulated in lung cancer [53], whereas the expression of *MANCR* is up-regulated in breast [54,55], gastric [56], and thyroid cancers [57]. Functionally, *MANCR* downregulates *miR-122a* in hepatocellular carcinoma [58], and a recent study shows that bromodomain protein 4 (BRD4), a member of the bromodomain and extra-terminal domain (BET) family of nuclear proteins, binds the super-enhancer region of *MANCR* locus, which in turn, influences the migration and invasion of prostate cancer cell line PC3 [59]. Mechanistically, *MANCR* acts as a miRNA sponge to sequester *miR-218*, which targets *RUNX2*, in mantle cell lymphoma [60]. In addition, *MIRLET7BHG* is shown to act as a miRNA sponge to sequester *miR-330-5p*, which targets the key signal transducer of the hedgehog signaling pathway, smoothened (*Smo*), in the exosomes of activated hepatic stellate cells [61]. However, none of these differentially expressed lncRNAs are studied in detail within the NAFLD context, which indicates the novelty in this topic.

### 2.3. Loss-of-Function Experiments to Uncover the Roles of Differentially Expressed LncRNAs

Since only a handful of lncRNAs are functionally characterized in the liver, especially related to NAFLD, we conducted loss-of-function experiments for one commonly up-regulated lncRNA, *LINC01639*. This lncRNA was chosen for further analysis because of its high expression levels in the human hepatocyte-derived carcinoma cell line, Huh-7, and the presence of only one transcript (thus, no isoforms for this lncRNA gene), according to the latest annotation provided by the Ensembl database.

When silenced in Huh-7 cells by siRNAs (Figure 4A), a differential expression of several genes was observed in *LINC01639* silenced cells compared to the control (cells treated with siRNA against scrambled sequence), including significant downregulation in the expression of NAFLD-linked (*TYMS*, *DUSP2*, *SRF*) TNF/TGF, and cytokine signaling (*CXCL3*, *TNFRSF10D*) and apoptosis-related genes (*BAD*), as well as growth factors *FGF21* and *IGF1* (Figure 4B).

To further understand the impact of silencing *LINC01639*, we developed a cellular model for NAFLD by treating Huh-7 cells with a mixture of fatty acids (FA) consisting of oleic and palmitic acids (Figure 4C). Interestingly, compared to the control, up-regulation of the cytokine signaling genes *CCL2* and *IL-6*, as well as the Rho GTPase *RAC2*, was recorded in *LINC01639* silenced cells (Figure 4D), suggesting that *LINC01639* might function in cytokine signaling.

### 2.4. A Web Database, LiverDB, for Exploration of NAFLD-Related Genes

To facilitate further understanding of lncRNAs in the liver and NAFLD, we built a web database, LiverDB, to catalog NAFLD-related genes (Figure 5A). LiverDB is an easy-to-use web database that displays expression changes of protein-coding and lncRNA genes in counts per million (CPM), reads per kilobase of transcript per million mapped reads (RPKM), and transcripts per kilobase million (TPM) (Figure 5B). Each study is linked to the data information provided by the GEO [62], and the hyperlink to GeneCards [63] is provided for each gene to obtain the known information related to the target gene. To enable exploration of differentially expressed genes, in the Explore tab of LiverDB, a volcano plot was generated (Figure 5C). In this volcano plot, the gene of interest selected from the Results Table on the left of the screen can be dynamically highlighted to visually inspect the ratio and FDR values of the target gene. These differentially expressed genes can be visually examined further by heatmaps (Figure 5D) and Kyoto Encyclopedia of Genes and Genomes (KEGG) pathway enrichment (Figure 5E). The displayed information should help to understand the genes and signaling pathways being affected by the two conditions being compared. Moreover, UpSet plots can be generated to display the numbers of differentially expressed genes (DEGs) that are found across study/contrast pairs in the Comparison tab (Figure 5F). Compared to a Venn diagram, an Upset plot provides a more efficient way of visualizing the intersections of all comparisons of the dataset, including in LiverDB. To allow further analysis, all data included in LiverDB can be downloaded as text files from the Download Table. In addition, the commands and nextflow [64] pipelines used for the content of LiverDB are available via the GitHub repository to allow for a further analysis of similar RNA-seq data.

## 3. Discussion

The salient findings of this study are: (1) many genes (both protein-coding and lncRNA genes) are differentially expressed in both subtypes of NAFLD patients compared to the healthy normal weight and obese individuals; (2) the overlap of differentially expressed genes between two independent studies are low, possibly due to the heterogeneity of affected cells in biopsy samples used for RNA-seq experiments; (3) the NAFLD-related lncRNA, *LINC01639*, is involved in the regulation of genes related to apoptosis, TNF/TGF and cytokine signaling, growth factors, as well as genes upregulated in NAFLD.

In this study, a large number of RNA-seq data were analyzed to interrogate NAFLD-related lncRNAs. Yet, there are several limitations to our study. First, all the RNA-seq data are from the strand-specific mRNA sequencing using the Illumina NextSeq 500 machine. Thus, lncRNAs without poly A tails were not detected in this study, which underestimates the impact of lncRNAs without poly A tails on the pathogenesis of NAFLD. Second, the RNA-seq data analysis pipeline used the annotation provided by the Ensembl database as a GTF file. Thus, only known and registered lncRNAs were analyzed in this study, which omits the novel lncRNAs. Third, all the RNA-seq data used in this study were generated from biopsy samples. Thus, the data variability might be high due to the heterogeneity of NALFD-affected livers [38,39]. To overcome this problem, single-cell RNA-seq (scRNA-seq) might be a solution; yet, only highly expressed lncRNAs without poly A tails can be recorded by the current method of scRNA-seq [65,66]. Fourth, only a molecular analysis of NAFLD-related lncRNA, *LINC01639*, was provided in this study. Thus, functional and mechanistic studies are needed to understand the contribution of *LINC01639* to the pathogenesis of NAFLD.

## 4. Materials and Methods

### 4.1. RNA-Seq Data Analysis

The following two datasets were used in this study: (1) GSE126848 [32] and (2) GSE135251 [21]. RNA-Seq data preprocessing pipeline was managed via Nextflow [64] (Version 21.10.6.5660). The raw data from the Sequence Read Archive (SRA) database was downloaded using the prefetch command within the SRA Toolkit (https://trace.ncbi.nlm.nih.gov/Traces/sra/sra.cgi?view=software (accessed on 17 February 2022) (Version 2.10.0). The raw data was then preprocessed by the fastq-dump command within the SRA Toolkit to generate fastq files. The fastq files were then preprocessed by fastp [67] (Version 0.23.2) to generate trimmed fastq files. The trimmed fastq files were aligned by STAR [68] (Version 2.7.10a) to align reads to the reference genome GRCh38.103 in order to generate raw read counts. From the raw read counts, the R packages, edgeR [69] (Version 3.38.1) and GenomicFeatures (Version 1.48.1) [70] were used to generate read count metrics (CPM, RPKM, TPM) and differentially expressed genes with fold change in the logarithmic of base two scale and FDR-adjusted *p*-values. Pathway analysis on select differentially expressed genes was conducted via Enrichr [71]. All codes are available at the LiverDB GitHub repository (https://github.com/Bishop-Laboratory/LiverDB (accessed on 23 June 2022) in the “preprocess” directory.

### 4.2. Data analysis and Visualization

Volcano plots were generated using the R-package ggplot2 [72]. Venn diagrams were plotted via http://bioinformatics.psb.ugent.be/webtools/Venn/ (accessed on 2 April 2022). To generate heat maps, MultiExperiment Viewer (MeV) [73] was used. The gene ontology terms were analyzed via the Database for Annotation, Visualization and Integrated Discovery (DAVID) v6.8 [74,75].

### 4.3. Interactive Web Database, LiverDB

The interactive web database LiverDB was built with R Shiny [76] (Version 1.7.1) and is hosted via shinyapps.io. The R packages used for plotting are ggplot2 [72] (Version 3.3.6) and ComplexHeatmap [77] (Version 2.12.0). For the displayed gene names, Ensembl IDs were mapped to the HUGO Gene Nomenclature Committee (HGNC) gene symbols via the R package biomaRt [78] (Version 2.52.0).

### 4.4. Cell Culture

Huh-7 cells were cultured in Minimum Essential Medium Eagle (Sigma-Aldrich, St. Louis, MO, USA, #M2279) supplemented with 10% fetal bovine serum (Sigma-Aldrich, #F4135), 1% MEM Non-essential Amino Acid Solution (Sigma-Aldrich, #M7145), 1% L-Glutamine solution (Sigma-Aldrich, #G7513), and 1% Penicillin-Streptomycin (Sigma-Aldrich, #P4333) at 37 °C with 5% CO_2_.

### 4.5. Transfection with siRNA and Treatment with Fatty Acids (FA)

The cells at 80% confluence were transfected with 50 nM of siRNA against *LINC01639* (sense: CAGACAUAGCAGGAUUUAA[dT][dT]/antisense: UUAAAUCCUGCUAUGUCUG[dT][dT]) using Lipofectamine RNAiMAX transfection reagent (Invitrogen, Waltham, MA, USA) according to the manufacturer’s protocol. Mission siRNA Universal Negative control #2 (Merck, Kenilworth, NJ, USA, SIC002) served as a control. The samples for RNA purification were collected 48 h after transfection. For development of NAFLD model, 24 h after the transfection, the growth media was exchanged to a starvation media (growth media without FBS) for another 24 h. Then, cells were treated with a mixture of fatty acids (FA) consisting of 2 mM oleic acid (OA; oleic acid-albumin from bovine serum, Merck O3008) and palmitic acid (PA; Sigma) conjugated to Bovine Serum Albumin (BSA) Fraction V, fatty acid free (Millipore, Burlington, MA, USA) in starvation media containing 10% BSA. Cells grown in starvation media containing 10% BSA served as a control. 48 h after treatment with FA cells were harvest for RNA isolation.

### 4.6. Lipid Accumulation Assay

Cells were seeded at density 1 × 10^4^ in a 96-well plate and grown overnight. After 24 h starvation, cells were treated with an FA mixture (2 mM OA, 2 mM PA, in 10% BSA media) or grown in 10% BSA media alone for 48 h. Wells without cells served as blank controls. Each treatment was performed in triplicates. The lipid accumulation was measured using Hepatic lipid accumulation/Steatosis assay kit (Abcam, Cambridge, UK, ab133131) according to the manufacturer’s protocol. Briefly, cells were fixed for 15 min, washed, and stained with Oil Red O solution for 20 min. The quantification of lipid accumulation was performed after dye extraction for 30 min and absorbance read at 490–520 nm.

### 4.7. Isolation of Total RNA and RT-PCR

RNeasy Mini Kit (Qiagen, Hilden, Germany, #74104) was used to isolate total RNA from cells and purified following the manufacturer’s protocol. SuperScript IV VILO Master Mix with ezDNase™ Enzyme (Thermo Fisher Scientific, Waltham, MA, USA, #11766500) was used to digest the genomic DNA and reverse transcribe one μg of total RNA for each sample to synthesize the first-strand complementary DNA (cDNA). After reverse transcription, the first-strand cDNA was diluted with DNase/RNase-free water to the concentration of 1 ng/μL. Quantitative reverse transcription polymerase chain reaction (qRT-PCR) reaction was performed with 1 ng of cDNA template per reaction using PowerUp SYBR Green Master Mix (Thermo Fisher Scientific, #A25777) via QuantStudio 6 Flex Real-Time PCR System (Thermo Fisher Scientific) with the annealing temperature at 60 °C. Relative fold expression was calculated by 2^-DDCt^ using ribosomal protein lateral stalk subunit P0 (*PRLP0*) as an internal control. The primer pairs were designed using Primer3 (http://bioinfo.ut.ee/primer3-0.4.0/ (accessed on 21 March 2022) [79] and in silico validated with the UCSC In-Silico PCR tool (https://genome.ucsc.edu/cgi-bin/hgPcr (accessed on 21 March 2022) before extensive testing by conventional RT-PCR reaction followed by running the PCR product on an agarose gel to examine for a single band of the expected size for each primer pair. The primer sequences are provided in Appendix A.

### 4.8. Statistics

Data are presented as mean ± S.E.M. unless otherwise indicated. Two-sample, two-tail, heteroscedastic Student’s *t*-test was performed to calculate a *p*-value via Microsoft Excel.

## Figures and Tables

**Figure 1 ncrna-08-00056-f001:**
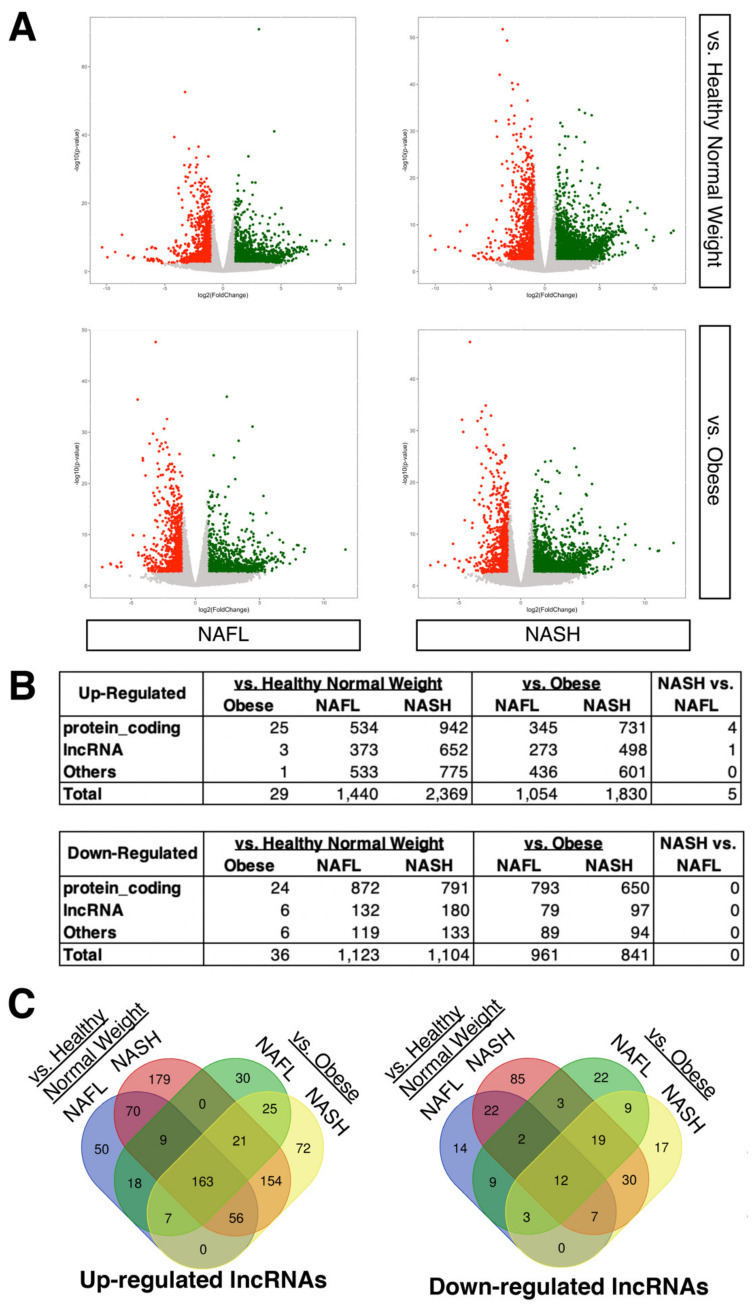
Gene expression profiling of NAFL and NASH patients compared to healthy normal weight and obese individuals. (**A**) Volcano plots comparing different conditions. The numbers of samples are NAFL (*n* = 15), NASH (*n* = 16), healthy normal weight (*n* = 14), and obese individuals (*n* = 12). Up-regulated genes are colored in red, and down-regulated genes are in green. (**B**) Tables of differentially expressed genes. The Others category includes any genes other than protein-coding or lncRNAs based on the biotypes provided by the Ensembl database. (**C**) Venn diagrams of up- and down-regulated lncRNAs.

**Figure 2 ncrna-08-00056-f002:**
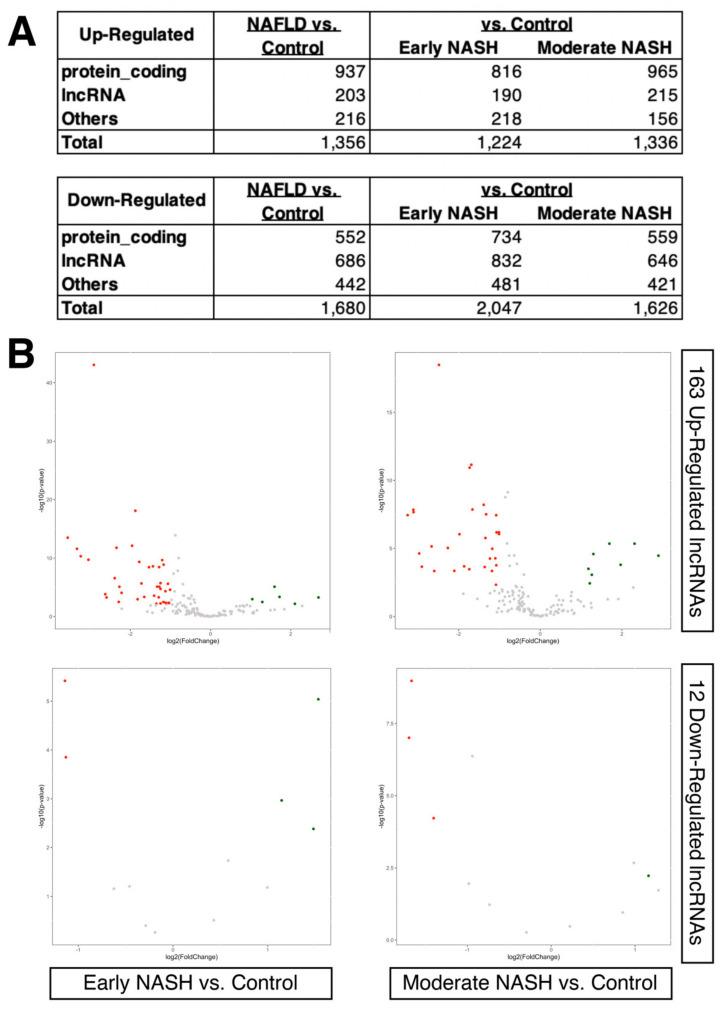
Gene expression changes in different stages of NASH compared to obese individuals. (**A**) Tables of differentially expressed genes. The Others category includes any genes other than protein-coding or lncRNAs based on the biotypes provided by the Ensembl database. (**B**) Volcano plot of early (*n* = 138) and moderate NASH patients (*n* = 68) compared to the control (obese individuals; *n* = 10). Only 163 up- and 12 down-regulated lncRNAs identified in the previous dataset (GEO accession number, GSE126848) are shown in each volcano plot, as indicated in the figure. Up-regulated genes are colored in red, and down-regulated genes are in green.

**Figure 3 ncrna-08-00056-f003:**
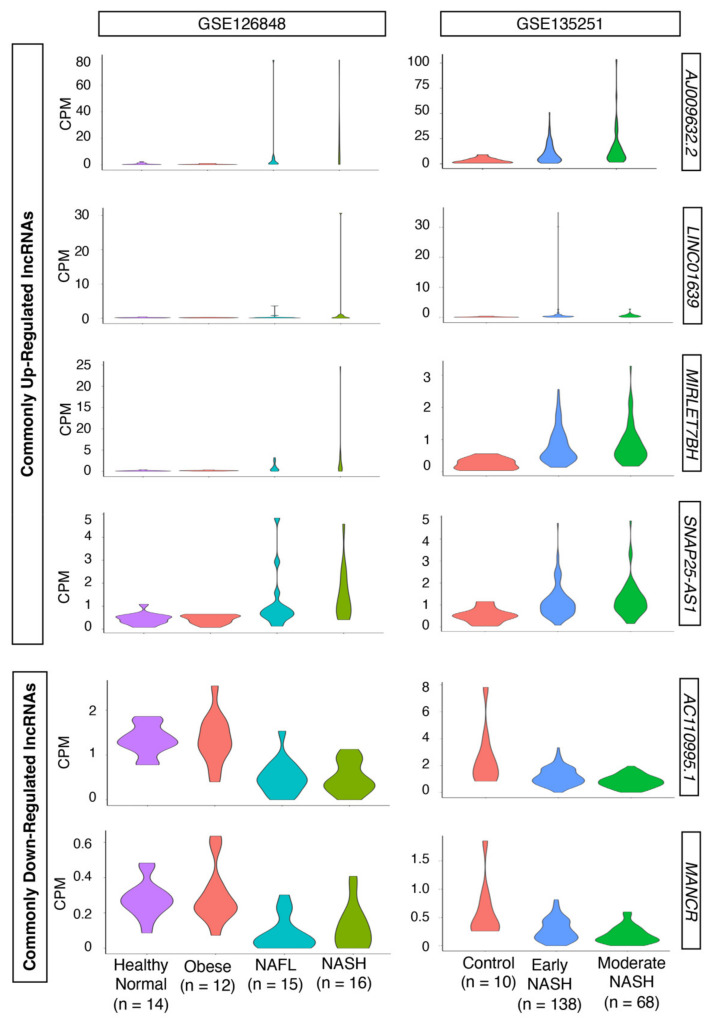
Expression profiles of commonly up- or down-regulated lncRNAs. Violin plots show counts per million (CPM) values of each sample group for each dataset, as indicated in the figure. The number of samples for each category is indicated in the figure.

**Figure 4 ncrna-08-00056-f004:**
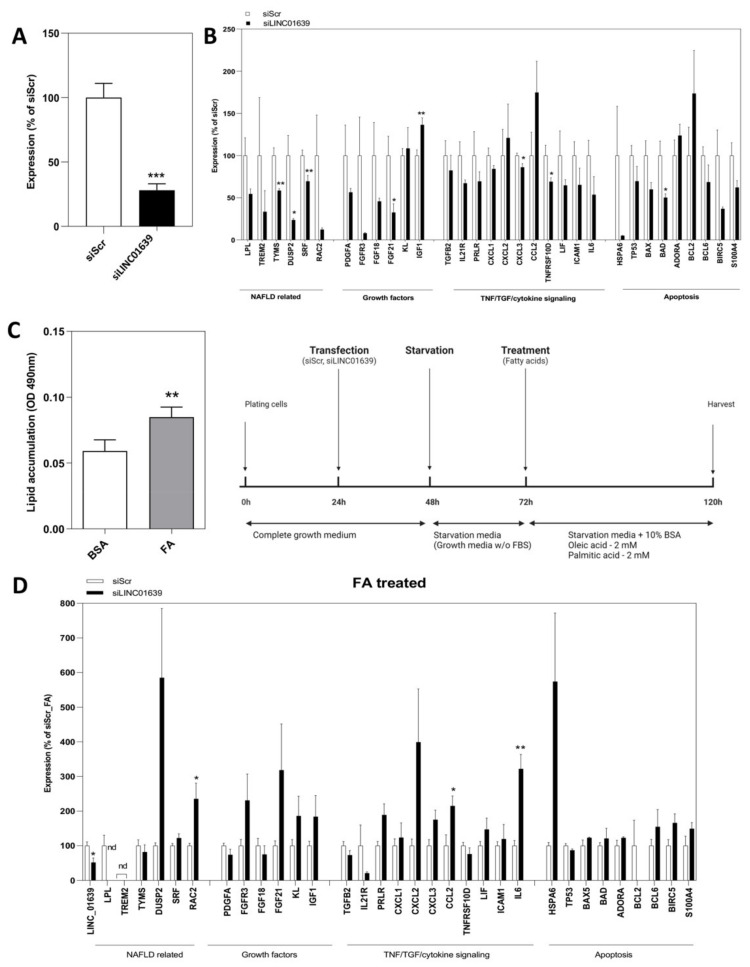
Role of *LINC01639* in regulation of gene expression in hepatocytes. (**A**) Silencing of *LINC01639* in Huh-7 cells, Error bars represent mean ± S.E.M. *** *p* < 0.001. (**B**) Gene expression profile after silencing of *LINC01639* in Huh-7 cells. *n* = three biological replicates. * *p* < 0.05; ** *p* < 0.01. (**C**) NAFLD cellular model. Huh-7 cells treated with a mixture of fatty acids (FA) consisting of 2 mM oleic acid and 2 mM palmitic acid at the ratio of 1:1. The schematic experimental timeline for NAFLD cellular model generation is presented. (**D**) Gene expression profile of NAFLD cellular model upon silencing of *LINC01639 n* = three biological replicates. * *p* < 0.05; ** *p* < 0.01. nd = not detected.

**Figure 5 ncrna-08-00056-f005:**
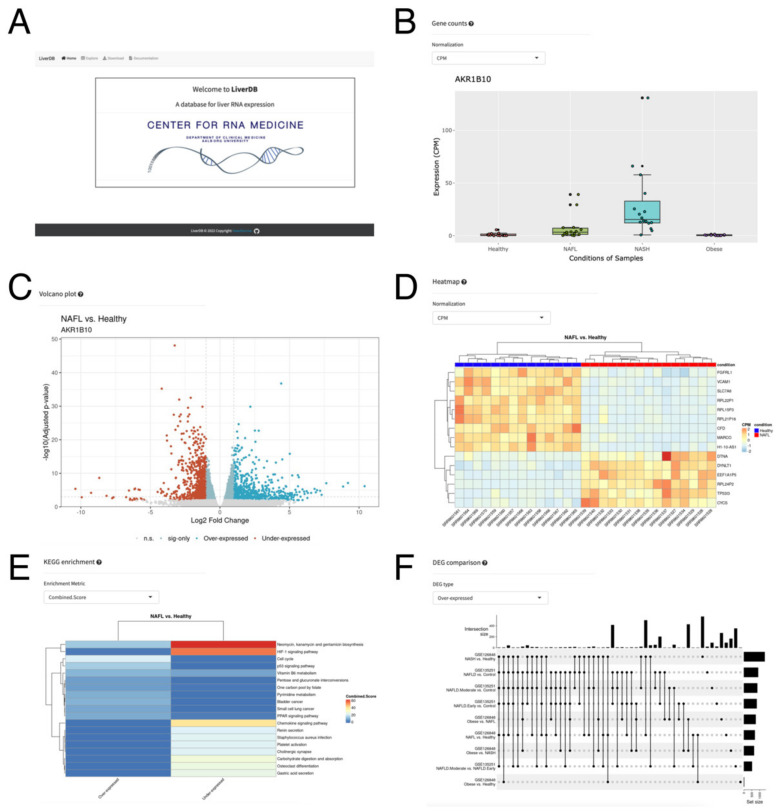
LiverDB. (**A**) The top page of LiverDB. (**B**) An interactive box plot showing the distribution of gene expression values across samples within each biological condition in the selected study. (**C**) A volcano plot of selected comparative conditions. (**D**) A heatmap of differentially expressed genes. (**E**) A pathway enrichment plot displaying the top results from KEGG pathway analysis as a heatmap. (**F**) An UpSet plot showing the number of differentially expressed genes that are found across study/contrast pairs.

## Data Availability

The Appendix A can be found on the GitHub repository: https://github.com/heartlncrna/Analysis_of_NAFLD_Studies (accessed on 23 June 2022). All codes used to generate LiverDB are available on the GitHub repository: https://github.com/Bishop-Laboratory/LiverDB (accessed on 23 June 2022).

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
