# Peer review of "Systematic Analysis of Long Non-Coding RNA Genes in Nonalcoholic Fatty Liver Disease"

_ncrna, 2022, doi:10.3390/ncrna8040056_

Round 1

Reviewer 1 Report

Review on the manuscript titled “LiverDB: Systematic analysis of long non-coding RNA genes in nonalcoholic fatty liver disease

                I can see the authors changing their accent from database to the rather thorough analysis they’ve performed. I sincerely believe  the study highlights the issues that would be quite actual  for the researchers in the field. The only note I’d like to propose is omitting ‘LiverDB’ from the title. I don’t downgrade the labor in creating the database and accept it importance, originality and relevance, but the data analysis per se conducted and described in the manuscript is of primary matter for the study objective. If the authors insist, they may leave it as is.

    Overall, the manuscript is well written and mature, the results look reliable and relevant. Again, some key gene networks outlining and elaborations would enhance the manuscript, but I acknowledge it may take some time to assemble, so not insisting on that.

Author Response

Thank you very much for your valuable comments. The word, LiverDB, has been removed from the title as suggested.

Reviewer 2 Report

A minor suggestion is to improve Figure 5 quality, making it more clear.

Author Response

Figure 5 has been modified to make the images clearer and in a higher resolution.

This manuscript is a resubmission of an earlier submission. The following is a list of the peer review reports and author responses from that submission.

Round 1

Reviewer 1 Report

In this study investigated the potential and functional long non-coding RNA genes in NAFLD. Overall, the study was well-performed using the database analysis and provide a website for researchers who are studying in this field. Only two minor suggestions, briefly give few examples of lncRNA in NAFLD and separate the number and unit in the methods (line 324). 

Reviewer 2 Report

Review on the article titled “LiverDB: Systematic analysis of long non-coding RNA genes in
nonalcoholic fatty liver disease” by Ilieva et al., 2022.

                The authors studied non-coding RNA in nonalcoholic fatty liver disease (NFALD), and steatohepatitis (NASH). For that they used the published RNA-Seq data (GSE126848; Suppli et al., 2019; PMID: 30653341) and GSE135251 (Govaere et al., 2020; PMID: 33762733; 33268509).

                After selecting and annotating ncRNA expressions profiles the author report that “Similar to protein-coding genes, many lncRNAs are dysregulated in NAFLD patients compared to healthy and obese individuals, suggesting that understanding the functions of dysregulated lncRNAs may shed light on the pathology of NAFLD. To demonstrate the functional importance of lncRNAs in the liver, loss-of-function experiments were performed for one NAFLD-related lncRNA, LINC01639, which showed that it is involved in the regulation of genes related to apoptosis, TNF/TGF, and cytokine signaling, and growth factors as well as genes upregulated in NAFLD” (abstract section).

                Thus, LINC01639 only was further elaborated by loss of function experiments. The database created by the authors yields no functional potential and is based on the data not generated by the authors themselves. If the db is the ultimate aim of the authors judging by outlining it in the title, they should illustrate its functional abilities that I cannot see herein. The only useful button is ‘download data’, and authors should prove it otherwise in the manuscript.

                Also, the major shortcoming of informative, biological part is the absence of particular gene networks with DEGs abundance and their analysis underscoring the basic processes affected in the pathologies.

                Overall, I can see no clear outcome from the article. Considering the citation rank of the journal, the manuscript current form lacks the outcome value needed for achieving it.

Notes:

1)      No exact references on the data used in the study is provided in M&M section.

2)      No reference on rather keystone paper published in Nature, 2021 on NASH data used by the authors. (PMID: 33762733).

3)      Supplementary tables contain DEGs in Pairwise comparisons, both coding and noncoding genes, of all groups (11 tables). The titles of the tables reside at the bottom (please move to the top). 2 tables (Table S13, S14) are just void, hopefully will be arranged.